

# Risk factors analysis and nomogram development for myelosuppression in diffuse large B-cell lymphoma patients undergoing first-line chemotherapy: a dual-centre retrospective cohort study

Xuexing Wang[1,2], Rong Zhang[3], Haoling Xie[1], Jinsong Xu[1], Jie Chu[4], Chunmei Wei[1] and Quanfang Chen[2]

[1] Department of Oncology, Anning First People's Hospital Affiliated to Kunming University of Science and Technology, Kunming, Yunnan, China
[2] Department of Pulmonary and Critical Care Medicine, The First Affiliated Hospital of Guangxi Medical University, Nanning, Guangxi Zhuang Autonomous Region, China
[3] Department of Geriatric Oncology, The Third Affiliated Hospital of Kunming Medical University, Kunming, Yunnan, China
[4] Department of Oncology, Ziyang Central Hospital, Ziyang, Sichuan, China

Corresponding author
Quanfang Chen,
chenquanfang555@163.com

## ABSTRACT

**Objective:** The primary objective of this research was to examine the characteristics of myelosuppression following first-line chemotherapy in patients suffering from diffuse large B-cell lymphoma (DLBCL). Furthermore, the study aimed to identify and analyze the risk factors impacting myelosuppression after chemotherapy and to construct a predictive model for evaluating the risk of myelosuppression.

**Methods:** This retrospective cohort study was conducted across two medical centers. The study included 243 patients with DLBCL treated at the Anning First People's Hospital Affiliated with Kunming University of Science and Technology from January 2022 to December 2023 as the development cohort, and 107 patients treated at the Third Affiliated Hospital of Kunming Medical University from January 2024 to May 2024 as the validation cohort. The study investigated the incidence of myelosuppression in all patients, identified independent factors influencing this condition through logistic regression analysis, and constructed and validated a nomogram. Finally, the model's performance was evaluated using both internal and external validation cohorts.

**Results:** The research rigorously incorporated a cohort of 243 DLBCL patients, with myelosuppression observed in 93 individuals (38.27%). Multifactorial analysis revealed that the chemotherapy cycle, age, Ann Arbor stage, surgical history, and neutrophil levels were independently correlated with myelosuppression following initial chemotherapy in DLBCL patients. A nomogram was developed based on the multifactorial analysis. The receiver operating characteristic (ROC) analysis revealed myelosuppression in the nomogram of both the development set (area under the curve (AUC = 0.834, 95% CI [0.785–0.884]) and the validation set (AUC = 0.861, 95% CI [0.791–0.931])), indicating clear differentiation. Further calibration curve analysis and decision curve analysis (DCA) revealed strong calibration and clinical utility of the column-line graph model.

**Conclusion:** Patients with DLBCL are at an increased risk and frequency of myelosuppression following first-line chemotherapy. The development of a highly accurate prediction model for myelosuppression in this patient population facilitates individualized treatment strategies. Future studies should focus on expanding the sample size and developing and validating the model in additional types of cancer.

# BACKGROUND

Lymphoma is a malignant neoplasm arising from the lymphohematopoietic system, comprising 3–4% of all malignant tumors globally (*Li et al., 2021*). It is currently ranked among the top ten most prevalent malignancies worldwide (*Wang et al., 2020*). The World Health Organization (WHO) categorizes lymphoma into Hodgkin's lymphoma (HL) and non-Hodgkin's lymphoma (NHL) (*Swerdlow et al., 2016*). In China, NHL represents over 85% of all lymphomas and approximately 4% of all malignant tumors (*Cai et al., 2021*). NHL is categorized based on cellular origin, with B lymphocytes (B cells) comprising approximately 86% of cases, and T cells and natural killer (NK) cells accounting for approximately 14% (*Al-Khreisat et al., 2022*). The most prevalent subtype of NHL is diffuse large B-cell lymphoma (DLBCL), representing approximately 30% to 40% of cases (*Berehab et al., 2021*). Lymphoma remains a significant public health issue, with its global burden steadily increasing. From 1990 to 2019, the age-standardized incidence rates of both HL and NHL rose, with NHL showing a particularly marked increase in prevalence, especially in middle sociodemographic index (SDI) regions. Although mortality rates have declined due to advancements in treatment, the overall disease burden, measured in disability-adjusted life years (DALYs), remains high, highlighting the need for improved prevention and management strategies (*Shen et al., 2024*). Cytotoxic chemotherapy remains the first line treatment for NHL (*Zhao et al., 2021*). However, chemotherapy is associated with a range of potent toxic side effects that prevent many patients from completing the treatment, potentially contributing to tumor progression and recurrence, prolonged hospitalization, increased financial strain, and even mortality (*Yang et al., 2023*; *Mayer et al., 2022*). These adverse effects primarily encompass myelosuppression, immunosuppression, gastrointestinal disturbances, hepatic and renal dysfunction, cardiotoxicity, and other complications, with myelosuppression being the most prevalent (*Xu et al., 2022*; *Gao et al., 2023*).

Nonetheless, a relatively small number of exploratory studies have investigated the risk factors for myelosuppression following chemotherapy, representing a notable gap in the literature. To our knowledge, no predictive modeling study has explored myelosuppression for first-line chemotherapy in DLBCL patients. Therefore, the present study aimed to address this gap and provide healthcare professionals with early identification of risk factors and effective prophylactic interventions. This retrospective cohort study was conducted utilizing data from 243 patients with DLBCL treated at the Anning First

People's Hospital Affiliated with Kunming University of Science and Technology between January 2022 and December 2023 as the development cohort, and 107 patients treated at the Third Affiliated Hospital of Kunming Medical University (Yunnan Cancer Hospital) between January 2024 and May 2024 as the prospective validation cohort. The cohort was analyzed and an effective prediction model was established, serving as a theoretical foundation for developing prevention strategies against myelosuppression in DLBCL patients undergoing chemotherapy.

## METHODS

### Study design and patient selection

This retrospective cohort study compiled and analyzed the medical records of 243 patients diagnosed with DLBCL who were hospitalized at the Anning First People's Hospital Affiliated with Kunming University of Science and Technology between January 2022 and December 2023, and 107 patients hospitalized at the Third Affiliated Hospital of Kunming Medical University (Yunnan Cancer Hospital) between January 2024 and May 2024. The inclusion criteria are listed as follows: (1) Patients diagnosed with DLBCL in accordance with the Chinese Guidelines for the Diagnosis and Treatment of DLBCL (2022 edition) based on histopathological, cytological, and imaging examinations and following the guidelines for standardized anti-tumor therapy; (2) Patients who have received at least one cycle of standardized chemotherapy according to the Chinese Society of Clinical Oncology Guidelines (CSCO) guidelines; (3) All patients performed complete blood count, liver and renal function, and electrolytes laboratory tests (both inpatient and outpatient) within 1 week before and after treatment. The exclusion criteria include the following: (1) Patients with a diagnosis of non-DLBCL; (2) Patients assessed as recurrent or refractory by the clinician; (3) Patients on the prophylactic use of drugs affecting blood counts prior to chemotherapy, such as granulocyte colony-stimulating factor (G-CSF), granulocyte-macrophage colony-stimulating factor (GM-CSF), erythropoietin (EPO), thrombopoietin (TPO), *etc*; (4) Patients with severe cardiac, pulmonary, or renal insufficiency during the period or patients whose treatment was interrupted by transfer to ICU or CCU during hospitalization; (5) Patients with incomplete medical records (patients with ≥20% missing). Figure 1 illustrates the experimental roadmap. Considering the retrospective nature of this study, the study protocol was approved by the Ethics Committee of the Anning First People's Hospital Affiliated with Kunming University of Science and Technology (approval number 2024-035 (Self-Horizontal)-01). Data analysis was conducted anonymously to safeguard patient privacy, thereby obviating the need for informed consent. The study adhered to the principles outlined in the Good Clinical Practice Guidelines (GCP) and the Declaration of Helsinki.

### Data collection

Comprehensive patient data was gathered *via* the hospital information system (HIS), including clinical profile (fundamental attributes, medical background, patient treatment characteristics) and laboratory metrics. The clinical profile of patients encompassed variables such as Ann Arbor stage, chemotherapy cycles, bone marrow invasion, age,

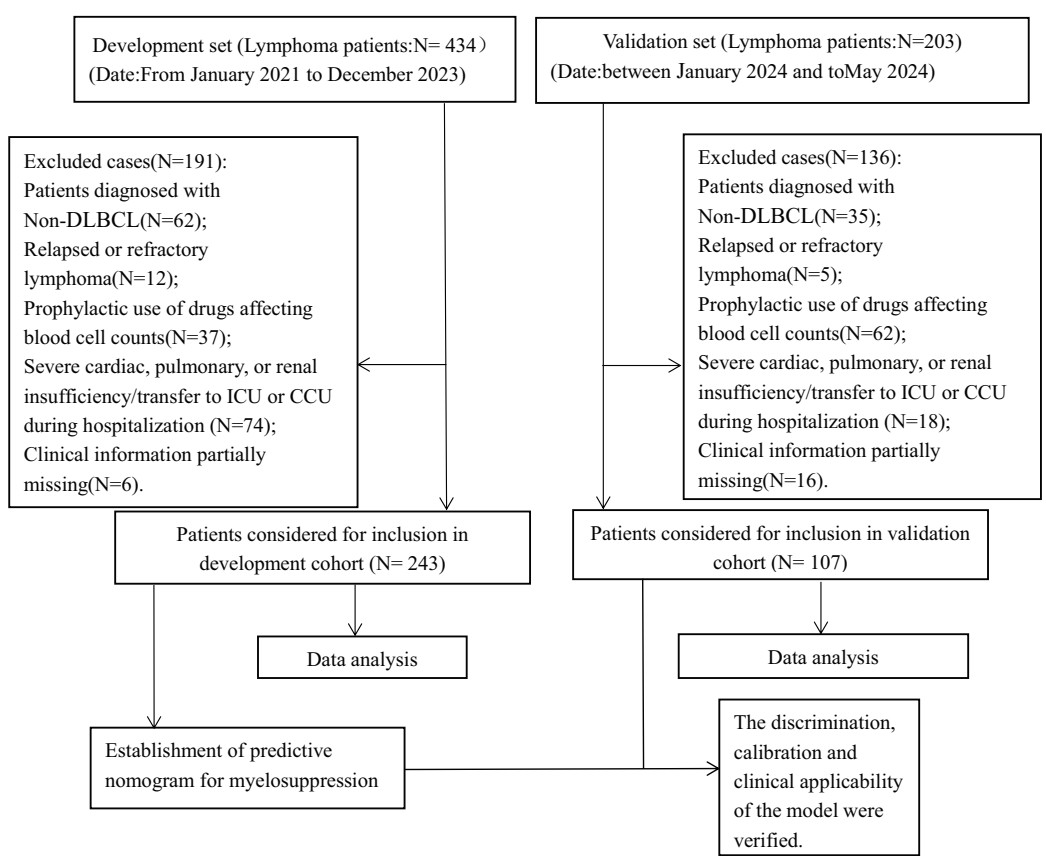

**Figure 1 Experimental roadmap of this study.** This experimental roadmap indicates the inclusion and exclusion for patients and the workflow of the method present in this study.

height, weight, smoking history, drinking history, surgical history (last 12 months), diabetes mellitus, hypertension, infectious conditions (within 1 month), and history of coronary artery disease. The laboratory metrics consisted of blood counts (including white blood cells, absolute neutrophils, hemoglobin, red blood cells, and platelet count) as well as biochemistry markers (such as albumin, globulin, alanine aminotransferase, aspartate aminotransferase, and creatinine). The clinical data was entered into Excel 2019 and thoroughly reviewed by two trained research team members. Missing data were handled using multiple imputations, as outlined in the statistical analysis section.

## Statistical analysis

Categorical data were expressed as numbers and percentages and were compared using chi-square tests. Numerical variables were tested for normality using the Shapiro–Wilk test. Normally distributed data were expressed as mean ± standard deviation (SD) and analyzed using independent samples t-tests. In contrast, data not conforming to a normal distribution were expressed as median and quartiles (M (Q1, Q3)) and compared using the rank sum test. The outcome variable in this study was myelosuppression, categorized according to the World Health Organization (WHO) chemotherapy toxicity criteria
(*World Health Organization, 2011*). Grades 0-I denoted the absence or mild presence of myelosuppression, while grades II–IV indicated moderate to severe myelosuppression. Patients falling within grades 0 were designated as the control group, whereas those falling within grades I–IV were classified as the case group, as per the criteria presented in Table 1. To identify potential predictors of myelosuppression, univariable logistic regression analysis was first performed. Variables with $P$-values < 0.05 were included in a multivariable logistic regression model using a stepwise forward selection approach. The statistically significant predictors retained in the multivariable analysis were used to construct a nomogram for individualized prediction of myelosuppression risk. To avoid overfitting, we applied the events per variable (EPV) principle, which recommends a minimum of 5–10 outcome events per predictor variable. In this study, 134 patients developed myelosuppression, allowing the safe inclusion of up to 13–26 variables. The discrimination, calibration, and clinical utility of the model were evaluated using the area under the ROC curve (AUC), calibration plots, and decision curve analysis (DCA), respectively. The AUCs of the development and validation sets were compared using the DeLong test to assess the statistical difference between the two ROC curves. All statistical analyses were performed separately using SPSS 26.0 (IBM, Armonk, NY, USA) and R software (version 4.2.0) to ensure clarity regarding the tools used in different analyses. In this study, $P < 0.05$ was considered statistically significant. For missing data, a multiple imputation method was applied to continuous variables, and a similar approach was used for categorical variables. Patients with missing data were excluded only if the missing values could not be imputed, ensuring that the imputation process did not introduce bias. Data removal occurred primarily when the missing values exceeded 30% for key variables, such as baseline clinical features. Most patients with DLBCL were treated with R-CHOP, CHOP, or R-CHOP-like regimens as first-line induction chemotherapy according to the CSCO guidelines. The clinical staging followed the modified Ann Arbor classification for non-Hodgkin lymphoma (NHL) (*Cheson et al., 2014*). A total of 243 patients from the Anning First People's Hospital Affiliated to Kunming University of Science and Technology were retrospectively included in the development cohort, and 107 patients from the Third Affiliated Hospital of Kunming Medical University were prospectively included in the validation cohort. The two groups were assigned with a 7:3 distribution to ensure sufficient sample sizes for both model development and validation.

## RESULTS

### Characterization of the study population and the occurrence of myelosuppression

In accordance with the established inclusion criteria, a total of 243 patients diagnosed with DLBCL at the First People's Hospital of Anning affiliated with Kunming University of Science and Technology between January 2022 and December 2023 were retrospectively selected as the development set. Subsequently, data from 107 patients with DLBCL treated at the Third Affiliated Hospital of Kunming Medical University from January 2024 to May 2024 were prospectively analyzed as the validation set. The distribution of cases between

**Table 1 Recommendation for grading of WHO criteria.**

| Items | 0 degree | I degree | II degree | III degree | IV degree |
|---|---|---|---|---|---|
| Leukocyte ($*10^9$/L) | ≥4.0 | 3.0–3.9 | 2.0–2.9 | 1.0–1.9 | <1.0 |
| Hemoglobin (g/L) | ≥110 | 95–109 | 80–94 | 65–79 | <65 |
| Platelets ($*10^9$/L) | ≥100 | 75–99 | 50–74 | 25–49 | <25 |
| Neutrophils ($*10^9$/L) | >2.0 | 1.5–1.9 | 1.0–1.4 | 0.5–0.9 | <0.5 |

the two centers was maintained at a ratio of 7:3. The demographic data, disease, treatment-related data, and laboratory data of both groups are displayed in Table 2. No statistically significant differences were observed between the development and validation sets in terms of gender, surgical history (within the last 12 months), chemotherapy cycles, BMI, smoking history, drinking history, diabetes mellitus, hemoglobin (HGB), aspartate aminotransferase (AST), and creatinine (Cr) (all $P > 0.05$). Table 3 illustrates the demographic and clinicopathologic features of the patients in the development set. In this study, the optimal cut-off values of the ROC curves were utilized to determine the critical values of patient age and BMI. The median age of the 243 DLBCL patients was 65 years (range: 50.50–77.00 years), including 114 females (46.91%) and 129 males (53.09%). The differences in baseline characteristics between the two groups of patients were compared. The results revealed statistically significant differences in age, Ann Arbor stage, surgical history, chemotherapy cycles, bone marrow invasion, red blood cells (RBC), white blood cells (WBC), HGB, neutrophils, and creatinine between the two groups ($P < 0.05$). As shown in Table 3 there are no significant statistical differences between the two groups of patients in terms of gender, *etc.* ($P > 0.05$). In the development set, myelosuppression was observed in 93 patients, representing an incidence of 38.27%, with the predominant myelosuppression grade being degree II. Similarly, in the validation set, myelosuppression was identified in 45 patients, resulting in an incidence rate of 42.10%, with degree II being the most prevalent myelosuppression grade. The myelosuppression grades and incidence rates of both groups are illustrated in Table 4. Statistical analysis revealed no significant difference between the two groups, as indicated by a $P$ value of more than 0.05.

## The potential risks and multifactorial analysis of myelosuppression in DLBCL patients

Table 5 presents the potential risk factors for myelosuppression in 243 patients with DLBCL following initial chemotherapy, as determined by univariate logistic regression analysis. The primary risk factors identified included the Ann Arbor stage, chemotherapy cycles, bone marrow invasion, age, body mass index (BMI), surgical history, WBC, neutrophils, and creatinine. Subsequent analysis utilized multivariate regression to identify independent factors influencing myelosuppression development, revealing that Ann Arbor stage, chemotherapy cycles, age, surgical history, and neutrophils were the predominant factors influencing myelosuppression occurrence, as displayed in Table 6.

**Table 2 Demographic and clinical characteristics among all patients.**

| Characteristic | Entire set[1] (n = 350) No. (%) | Development set[1] (n = 243) No. (%) | Validation set[1] (n = 107) No. (%) | P[2] |
|---|---|---|---|---|
| Age, median (IQR) | 62.00 [48.00, 72.00] | 65.00 [50.50, 77.00] | 56.00 [43.50, 64.00] | <0.001 |
| Gender | | | | 0.294 |
| Male | 193 (55.14%) | 129 (53.09%) | 64 (59.81%) | |
| Female | 157 (44.86%) | 114 (46.91%) | 43 (40.19%) | |
| Ann Arbor stage | | | | 0.004 |
| I | 29 (8.29%) | 19 (7.82%) | 10 (9.35%) | |
| II | 94 (26.86%) | 64 (26.34%) | 30 (28.04%) | |
| III | 85 (24.29%) | 72 (29.63%) | 13 (12.15%) | |
| IV | 142 (40.57%) | 88 (36.21%) | 54 (50.47%) | |
| Surgical history (within the last 12 months) | | | | 0.706 |
| No | 282 (80.57%) | 194 (79.84%) | 88 (82.24%) | |
| Yes | 68 (19.43%) | 49 (20.16%) | 19 (17.76%) | |
| Chemotherapy cycles | | | | 0.956 |
| 1–3 | 210 (60.00%) | 146 (60.08%) | 64 (59.81%) | |
| 4–6 | 70 (20.00%) | 50 (20.58%) | 20 (18.69%) | |
| 7–9 | 45 (12.86%) | 30 (12.35%) | 15 (14.02%) | |
| >9 | 25 (7.14%) | 17 (7.00%) | 8 (7.48%) | |
| Bone marrow invasion | | | | 0.001 |
| No | 291 (83.14%) | 213 (87.65%) | 78 (72.90%) | |
| Yes | 59 (16.86%) | 30 (12.35%) | 29 (27.10%) | |
| BMI (kg/m$^2$), median (IQR) | 21.85 [19.56, 23.58] | 21.83 [19.70, 23.67] | 21.87 [19.27, 23.51] | 0.813 |
| Smoking history | | | | 0.427 |
| No | 284 (81.14%) | 194 (79.84%) | 90 (84.11%) | |
| Yes | 66 (18.86%) | 49 (20.16%) | 17 (15.89%) | |
| Drinking history | | | | 0.929 |
| No | 305 (87.14%) | 211 (86.83%) | 94 (87.85%) | |
| Yes | 45 (12.86%) | 32 (13.17%) | 13 (12.15%) | |
| Co-infection (within 1 month) | | | | <0.001 |
| No | 315 (90.00%) | 208 (85.60%) | 107 (100.00%) | |
| Yes | 35 (10.00%) | 35 (14.40%) | 0 (0.00%) | |
| Coronary artery disease | | | | 0.004 |
| No | 322 (92.00%) | 217 (89.30%) | 105 (98.13%) | |
| Yes | 28 (8.00%) | 26 (10.70%) | 2 (1.87%) | |
| Hypertension | | | | <0.001 |
| No | 253 (72.29%) | 162 (66.67%) | 91 (85.05%) | |
| Yes | 97 (27.71%) | 81 (33.33%) | 16 (14.95%) | |
| Diabetes mellitus | | | | >0.999 |
| No | 333 (95.14%) | 231 (95.06%) | 102 (95.33%) | |
| Yes | 17 (4.86%) | 12 (4.94%) | 5 (4.67%) | |
| Laboratory data | | | | |
| WBC (10$^9$/l), median (IQR) | 5.60 [4.11, 7.62] | 5.25 [3.78, 7.16] | 6.45 [4.95, 8.54] | <0.001 |

(Continued)

| Characteristic | Entire set[1] (n = 350) No. (%) | Development set[1] (n = 243) No. (%) | Validation set[1] (n = 107) No. (%) | P[2] |
|---|---|---|---|---|
| PLT ($10^9$/l), median (IQR) | 219.00 [165.25, 288.00] | 204.00 [156.50, 271.00] | 251.00 [190.00, 309.00] | 0.001 |
| HGB (g/l), median (IQR) | 132.00 [117.00, 144.00] | 133.00 [117.50, 143.50] | 130.00 [115.00, 146.00] | 0.883 |
| Neutrophils ($10^9$/L), median (IQR) | 3.56 [2.45, 4.79] | 3.29 [2.23, 4.58] | 4.06 [3.03, 5.22] | <0.001 |
| ALT (g/L), median (IQR) | 18.00 [13.00, 27.00] | 20.00 [14.00, 29.00] | 15.00 [11.00, 21.00] | 0.001 |
| AST (g/L), median (IQR) | 22.00 [17.00, 28.45] | 22.00 [18.00, 29.00] | 21.00 [16.00, 26.00] | 0.090 |
| Albumin (g/L), median (IQR) | 41.00 [37.00, 45.00] | 40.50 [36.95, 44.00] | 43.00 [38.50, 46.00] | <0.001 |
| Globulin (g/L), median (IQR) | 26.00 [23.00, 30.00] | 25.70 [22.70, 29.05] | 27.00 [24.00, 31.00] | 0.004 |
| Creatinine (mmol/l), median (IQR) | 72.00 [57.00, 84.00] | 72.00 [55.00, 84.00] | 73.00 [60.00, 84.20] | 0.207 |

Notes:
[1]: Median (IQR); n (%).
[2]: Wilcoxon rank sum test; Pearson's Chi-squared test; Fisher's exact test.

**Table 3 Demographic and clinicopathological characteristics of patients in the development set.**

| Characteristic | Overall[1] (N = 243[1]) | Control group[1] (n = 150) | Case group[1] (n = 93) | P[2] |
|---|---|---|---|---|
| Demographic characteristics | | | | |
| Age, median (IQR) | 65.00 [50.50, 77.00] | 62.00 [45.00, 73.00] | 66.00 [51.00, 77.75] | 0.006 |
| Gender | | | | 0.198 |
| Male | 129 (53.09%) | 85 (56.67%) | 44 (47.31%) | |
| Female | 114 (46.91%) | 65 (43.33%) | 49 (52.69%) | |
| Ann Arbor stage | | | | 0.020 |
| I | 19 (7.82%) | 15 (10.00%) | 4 (4.30%) | |
| II | 64 (26.34%) | 47 (31.33%) | 17 (18.28%) | |
| III | 72 (29.63%) | 42 (28.00%) | 30 (32.26%) | |
| IV | 88 (36.21%) | 46 (30.67%) | 42 (45.16%) | |
| Surgical history (within the last 12 months) | | | | 0.004 |
| No | 194 (79.84%) | 129 (86.00%) | 65 (69.89%) | |
| Yes | 49 (20.16%) | 21 (14.00%) | 28 (30.11%) | |
| Chemotherapy cycles | | | | <0.001 |
| 1–3 | 146 (60.08%) | 100 (66.67%) | 46 (49.46%) | |
| 4–6 | 50 (20.58%) | 18 (12.00%) | 32 (34.41%) | |
| 7–9 | 30 (12.35%) | 19 (12.67%) | 11 (11.83%) | |
| >9 | 17 (7.00%) | 13 (8.67%) | 4 (4.30%) | |
| Bone marrow invasion | | | | 0.016 |
| No | 213 (87.65%) | 138 (92.00%) | 75 (80.65%) | |
| Yes | 30 (12.35%) | 12 (8.00%) | 18 (19.35%) | |
| BMI (kg/m$^2$), median (IQR) | 21.83 [19.70, 23.67] | 21.97 [19.75, 23.70] | 21.64 [19.61, 23.51] | 0.595 |
| Smoking history | | | | 0.285 |
| No | 194 (79.84%) | 116 (77.33%) | 78 (83.87%) | |
| Yes | 49 (20.16%) | 34 (22.67%) | 15 (16.13%) | |
| Drinking history | | | | 0.771 |
| No | 211 (86.83%) | 129 (86.00%) | 82 (88.17%) | |

| Characteristic | Overall[1] (N = 243[1]) | Control group[1] (n = 150) | Case group[1] (n = 93) | P[2] |
|---|---|---|---|---|
| Yes | 32 (13.17%) | 21 (14.00%) | 11 (11.83%) | |
| Co-infection (within 1 month) | | | | 0.243 |
| No | 208 (85.60%) | 132 (88.00%) | 76 (81.72%) | |
| Yes | 35 (14.40%) | 18 (12.00%) | 17 (18.28%) | |
| Coronary artery disease | | | | 0.141 |
| No | 217 (89.30%) | 130 (86.67%) | 87 (93.55%) | |
| Yes | 26 (10.70%) | 20 (13.33%) | 6 (6.45%) | |
| Hypertension | | | | 0.124 |
| No | 162 (66.67%) | 94 (62.67%) | 68 (73.12%) | |
| Yes | 81 (33.33%) | 56 (37.33%) | 25 (26.88%) | |
| Diabetes mellitus | | | | 0.580 |
| No | 231 (95.06%) | 144 (96.00%) | 87 (93.55%) | |
| Yes | 12 (4.94%) | 6 (4.00%) | 6 (6.45%) | |
| Laboratory data | | | | |
| WBC ($10^9$/l), median (IQR) | 5.25 [3.78, 7.16] | 5.54 [4.27, 7.59] | 4.45 [3.46, 6.43] | <0.001 |
| PLT ($10^9$/l), median (IQR) | 204.00 [156.50, 271.00] | 210.00 [157.75, 272.50] | 197.00 [153.00, 266.00] | 0.405 |
| HGB (g/l), median (IQR) | 133.00 [117.50, 143.50] | 135.00 [119.25, 145.75] | 128.00 [115.00, 139.00] | 0.013 |
| Neutrophils ($10^9$/L), median (IQR) | 3.29 [2.23, 4.58] | 3.72 [2.62, 4.90] | 2.64 [1.86, 3.86] | <0.001 |
| ALT (g/L), median (IQR) | 20.00 [14.00, 29.00] | 20.00 [15.00, 28.75] | 20.00 [13.00, 29.00] | 0.965 |
| AST (g/L), median (IQR) | 22.00 [18.00, 29.00] | 22.00 [18.00, 27.75] | 23.00 [19.00, 30.00] | 0.197 |
| Albumin (g/L), median (IQR) | 40.50 [36.95, 44.00] | 40.05 [35.88, 44.00] | 40.60 [38.00, 43.00] | 0.909 |
| Globulin (g/L), median (IQR) | 25.70 [22.70, 29.05] | 26.10 [23.00, 29.85] | 24.40 [22.00, 29.00] | 0.072 |
| Creatinine (mmol/l), median (IQR) | 72.00 [55.00, 84.00] | 75.00 [57.00, 85.10] | 68.00 [52.00, 79.00] | 0.033 |

Notes:
[1]: Median (IQR); n (%).
[2]: Wilcoxon rank sum test; Pearson's Chi-squared test; Fisher's exact test.

**Table 4 Incidence and grading of myelosuppression after first-line chemotherapy in patients with DLBCL in the development and validation sets.**

| Groups (n; %) | 0 degree (n; %) | I degree (n; %) | II degree (n; %) | III degree (n; %) | IV degree (n; %) |
|---|---|---|---|---|---|
| Development set (243; 100%) | 150; (61.73) | 15 (6.17) | 52 (21.40) | 18 (7.40) | 8 (3.30) |
| Validation set (107; 100%) | 62; (57.94) | 10 (9.35) | 23 (21.50) | 8 (7.48) | 4 (3.74) |

Note:
P value < 0.05 for comparison of incidence and fractionation of myelosuppression in both groups.

## Establishment and validation of the nomogram for myelosuppression in DLBCL patients

A binary logistic model was utilized to construct the nomogram, incorporating the five identified significant factors. All of these factors were found to be significant predictors of myelosuppression formation within the logistic model. The resulting nomograms indicated that neutrophils had the greatest impact (OR = 4.838, 95% CI [2.041–11.469]), followed by surgical history, age, chemotherapy cycles, and Ann Arbor stage. The legend

**Table 5 Univariate logistic regression analysis in the development set.**

| Characteristics | OR (95% CI) | P |
| --- | --- | --- |
| **Ann Arbor stage** | | |
| I | Reference | |
| II | 1.356 [0.395–4.662] | 0.628 |
| III | 2.679 [0.808–8.878] | 0.107 |
| IV | 3.424 [1.053–11.138] | 0.041 |
| **Chemotherapy cycles** | | |
| 1–3 | Reference | |
| 4–6 | 3.865 [1.968–7.589] | <0.001 |
| 7–9 | 1.259 [0.554–2.859] | 0.583 |
| >9 | 0.669 [0.207–2.163] | 0.502 |
| **Bone marrow invasion** | | |
| No | Reference | |
| Yes | 2.760 [1.262–6.037] | 0.011 |
| **Age (years)** | | |
| ≥60 | Reference | |
| <60 | 0.566 [0.334–0.959] | 0.035 |
| **BMI (kg/m$^2$)** | | |
| <25.0 | Reference | |
| ≥25.0 | 0.388 [0.161–0.935] | 0.035 |
| **Surgical history (within the last 12 months)** | | |
| No | Reference | |
| Yes | 2.646 [1.396–5.016] | 0.003 |
| **WBC*** | | |
| Normal | Reference | |
| Decreased | 3.010 [1.681–5.392] | <0.001 |
| **Neutrophils*** | | |
| Normal | Reference | |
| Decreased | 4.228 [2.003–8.924] | <0.001 |
| **Creatinine (mmol/l)** | | |
| <73 | Reference | |
| ≥73 | 0.557 [0.329–0.944] | 0.030 |

Note:
WBC*: white blood cell, Normal: >4.0*10$^9$/L, Decreased: 3.0–4.0*10$^9$/L; Neutrophils*: Normal: >2.0*10$^9$/L, Decreased: <2.0*10$^9$/L.

accompanying Fig. 2 provides a comprehensive explanation of the nomogram's usage. The prediction model yielded a positive prediction probability threshold of 0.33, a sensitivity of 0.89, and a specificity of 0.61 (Youden value = 0.50). The calibration of the prediction model was assessed using the Hosmer–Lemeshow goodness-of-fit test, revealing a Hosmer–Lemeshow $\chi^2$ = 5.139, $P$ = 0.658. Additionally, the internally validated Bootstrap validation method yielded a C-index of 0.834, indicating the efficacy of the established prediction model. Furthermore, ROC analysis was performed to compare the predictive capabilities of the myelosuppressive nomogram models. The receiver operating

**Table 6 Multivariable logistic regression analysis in the development set.**

| Characteristics | OR (95% CI) | P |
|---|---|---|
| **Ann Arbor stage** | | |
| I | Reference | |
| II | 1.818 [0.468–7.062] | 0.388 |
| III | 3.332 [0.853–13.019] | 0.083 |
| IV | 4.324 [1.194–15.663] | 0.026 |
| **Chemotherapy cycles** | | |
| 1–3 | Reference | |
| 4–6 | 4.540 [2.066–9.976] | <0.001 |
| 7–9 | 1.713 [0.670–4.444] | 0.261 |
| >9 | 0.861 [0.229–3.242] | 0.825 |
| **Age (years)** | | |
| ≥60 | Reference | |
| <60 | 0.384 [0.196–0.751] | 0.005 |
| **Surgical history (within the last 12 months)** | | |
| No | Reference | |
| Yes | 2.775 [1.322–5.822] | 0.007 |
| **Neutrophils ($10^9$/L)** | | |
| >2.0 | Reference | |
| ≤2.0 | 4.838 [2.041–11.469] | <0.001 |

characteristic (ROC) curves of the predicted nomograms for both the development set (Fig. 3A) and the validation set (Fig. 3B) demonstrated area under the curve (AUC) values of 0.834 (95% CI [0.785–0.884]) and 0.861 (95% CI [0.791–0.931]), respectively, with a non-significant P-value of >0.05. The nomograms demonstrated excellent calibration in both the development and validation sets, suggesting strong discriminatory ability (Figs. 4A and 4B). Decision curve analyses were conducted on the predictive column line plots in both sets (Figs. 5A and 5B). A threshold probability range of 0–0.7 was identified as optimal for predicting myelosuppression formation using these column line plots. The findings from the DCAs indicate that predictive column-line plots provided a superior prediction of myelosuppression formation compared to categorizing all patients as either low or high risk. These results suggest that the column-line diagram prediction model has better clinical prediction and application value.

## DISCUSSION

The large majority of B-cell non-Hodgkin's lymphomas express the CD20 antigen, which has led to the adoption of rituximab in conjunction with the CHOP regimen as the primary treatment protocol for this type of lymphoma (Li et al., 2020). Chemotherapy, targeted therapy, and radiotherapy are recognized as the primary therapeutic modalities for DLBCL, each offering the potential to improve clinical outcomes for patients (Valls et al., 2016). However, myelosuppression represents a major adverse effect of chemotherapy in patients with non-Hodgkin lymphoma, leading to an elevated risk of

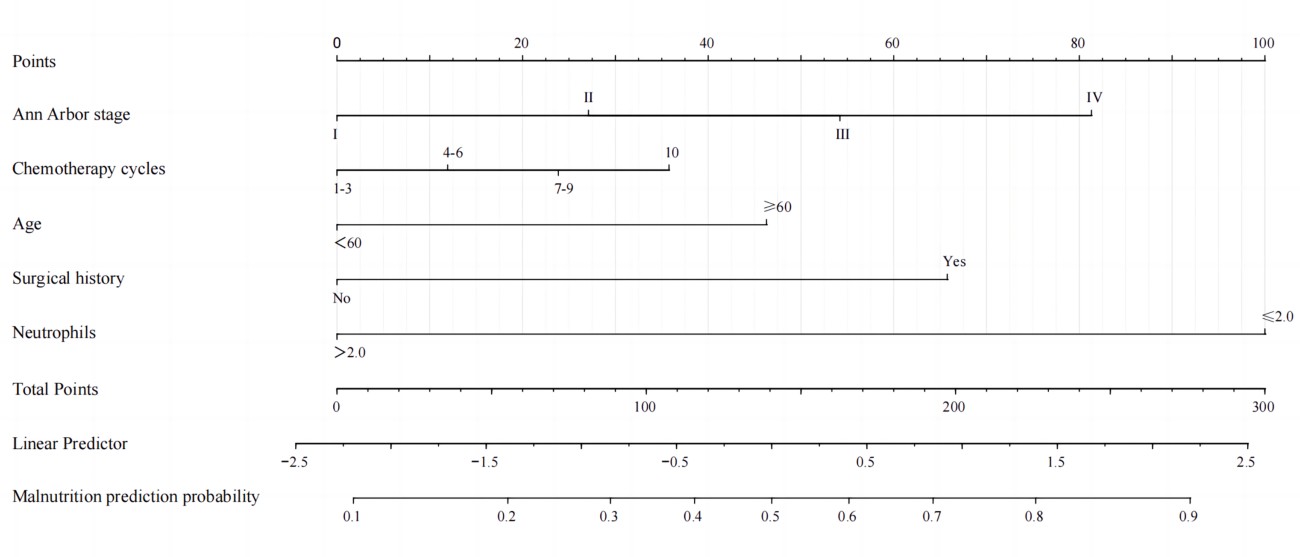

**Figure 2 A nomogram for predicting myelosuppression in DLBCL patients following the first-line chemotherapy.** A nomogram was developed to predict severe myelosuppression in DLBCL patients undergoing first-line chemotherapy, based on five predictors. To utilize the nomogram, each variable classification is plotted upwards on the "Points" axis to determine its corresponding points. The total points are then calculated on the "Total Points" axis, and a line is drawn downwards to the "Myelosuppression" axis to ascertain the probability of myelosuppression.

infection and hemorrhage due to variations in the half-life of granulocytes, platelets, and erythrocytes (*Yang, Zhong & Huang, 2023*). In clinical settings, clinicians frequently adjust the dosage or intervals of chemotherapeutic drugs to mitigate myelosuppression. However, these adjustments can impede treatment adherence, diminish therapeutic outcomes, prolong hospitalization, and impose a larger financial strain on patients (*Li et al., 2022*; *Bishnoi, Bennett & Reisman, 2017*). Therefore, anticipating the risk of myelosuppression and implementing prophylactic measures to elevate white blood cell and platelet counts, mitigate damage to the hematopoietic system from immunochemotherapy, and protect bone marrow hematopoietic function are essential strategies for ensuring successful treatment outcomes. Considering the scarcity of studies exploring the occurrence of myelosuppression after first-line chemotherapy in patients with DLBCL and the lack of evidence on the factors influencing its occurrence, the present study filled this gap in current national and international research, involving samples from two centers in southwestern China.

The data obtained from the two oncology centers included in this study revealed that the prevalence of myelosuppression among DLBCL patients was 38.27% (93 out of 243) and 42.10% (45 out of 107), respectively. The most common grade of myelosuppression in both groups was grade II, with frequencies of 21.40% and 21.50%, respectively. Notably, both the incidence and grade of myelosuppression between the two groups were highly

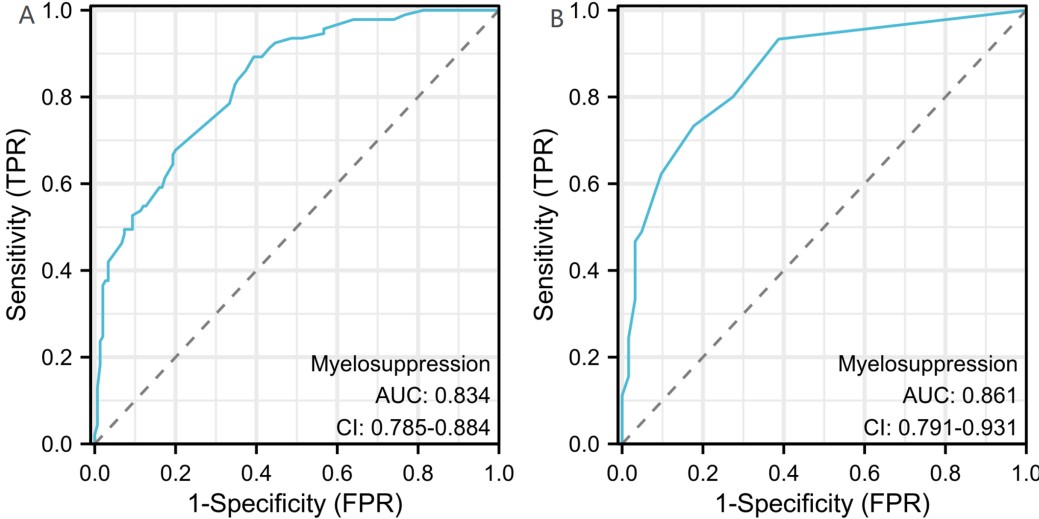

**Figure 3 Receiver operating characteristic curves (ROC) for the development and validation sets.** The 95% CI and *P*-value for the area under the ROC curve for the development set (A) and validation set (B) were 0.834 (95% CI [0.785–0.884]) and 0.861 (95% CI [0.791–0.931]), respectively. No significant difference was observed between the two AUCs (DeLong test, *P* > 0.05).

similar. The current study revealed a lower incidence of myelosuppression in patients with non-Hodgkin lymphoma compared to previous studies, which reported an incidence of around 50% in patients undergoing chemotherapy for solid tumors (*Zhang et al., 2021*; *Dong et al., 2023*; *Kiba et al., 2016*). Nonetheless, the causes of this discrepancy remain unclear but may be attributed to variations in the biological characteristics of lymphomas and solid tumors. For instance, differences in immunological responses can impact the choice of chemotherapy regimens for patients with lymphomas compared to solid tumors. Lymphomas are predominantly malignant tumors of immune cell origin and may exhibit lower levels of immunosuppression compared to solid tumors, potentially reducing the risk of myelosuppression. Moreover, variances in the chemotherapeutic agents and treatment protocols utilized for each type of cancer may also affect the incidence of myelosuppression. Patients with solid tumors may be prescribed anthracyclines such as cyclophosphamide, pegylated cyclophosphamide, and doxorubicin in their chemotherapy regimen. Other classes of chemotherapeutic agents, including antimetabolites like fluorouracil, methoxyaminothymidine, and cytarabine, or anti-tumor antibiotics such as adriamycin, etoposide, and doxorubicin, as well as DNA enzyme inhibitors, are also commonly utilized in cancer treatment protocols. Additionally, lymphomas exhibit a faster growth rate and higher sensitivity to chemotherapy, which may contribute to a more favorable response and reduced risk of myelosuppression in this patient population. In summary, individuals diagnosed with DLBCL lymphoma may exhibit more favorable baseline health, immune system functionality, and treatment tolerance compared to those with solid tumors, thereby potentially decreasing the risk of myelosuppression.

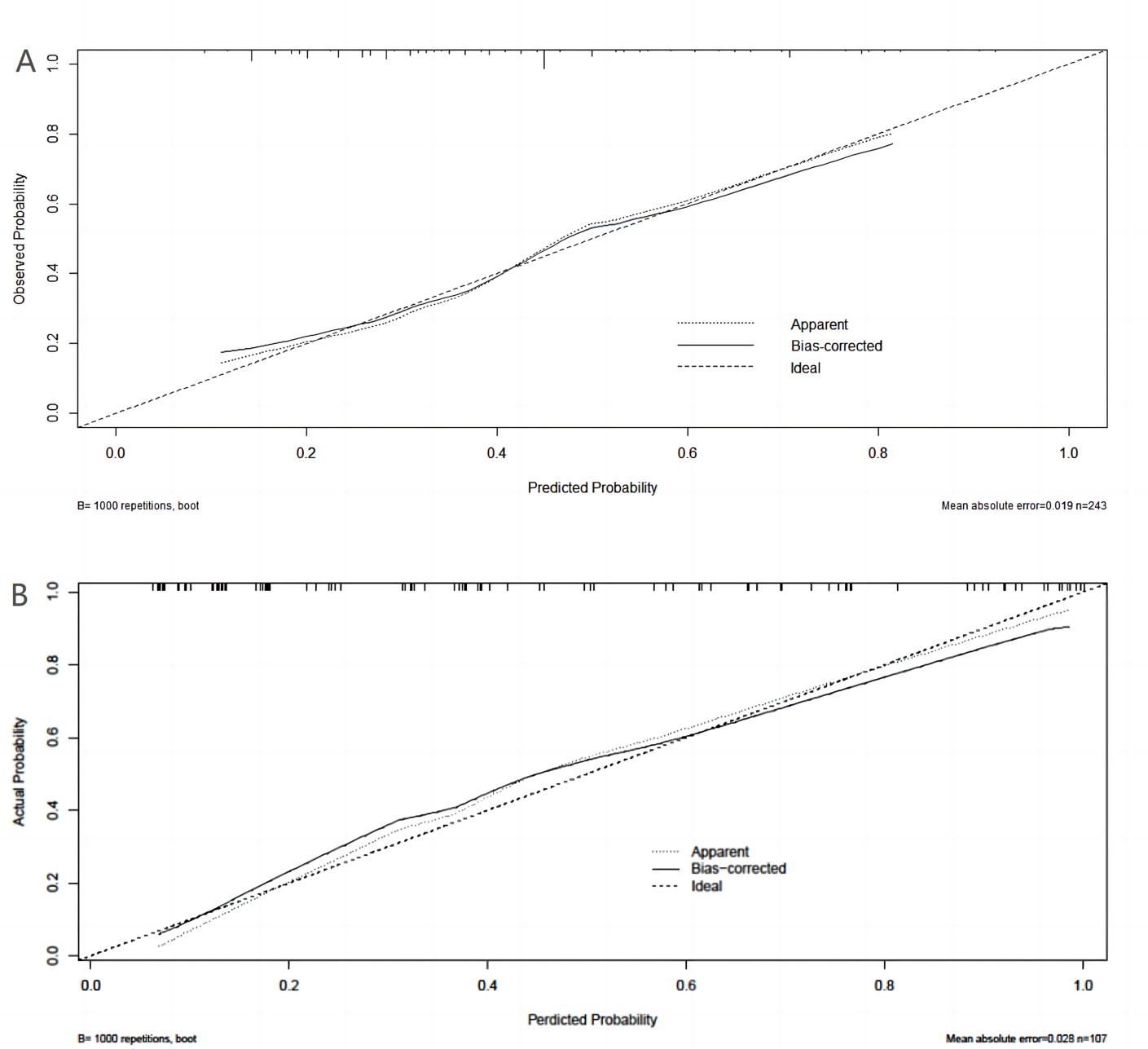

**Figure 4 Calibration curves for development set and validation set.** (A) Calibration curves for the development set. (B) Calibration curve for validation set. The y-axis represents the actual probability of myelosuppression occurring. x-axis represents the predicted probability of myelosuppression occurring. Diagonal: The black dashed line represents a perfect prediction using the ideal model. The gray dashed line represents the target parameter and the black solid line represents the performance of the model. The closer to the diagonal thick gray line represents a better prediction. Use bootstrap resampling (times = 1,000).

A binary logistic regression analysis was conducted to analyze the risk factors associated with myelosuppression in a cohort of 243 DLBCL patients undergoing first-line chemotherapy treatment. The analysis incorporated 22 clinically available and commonly

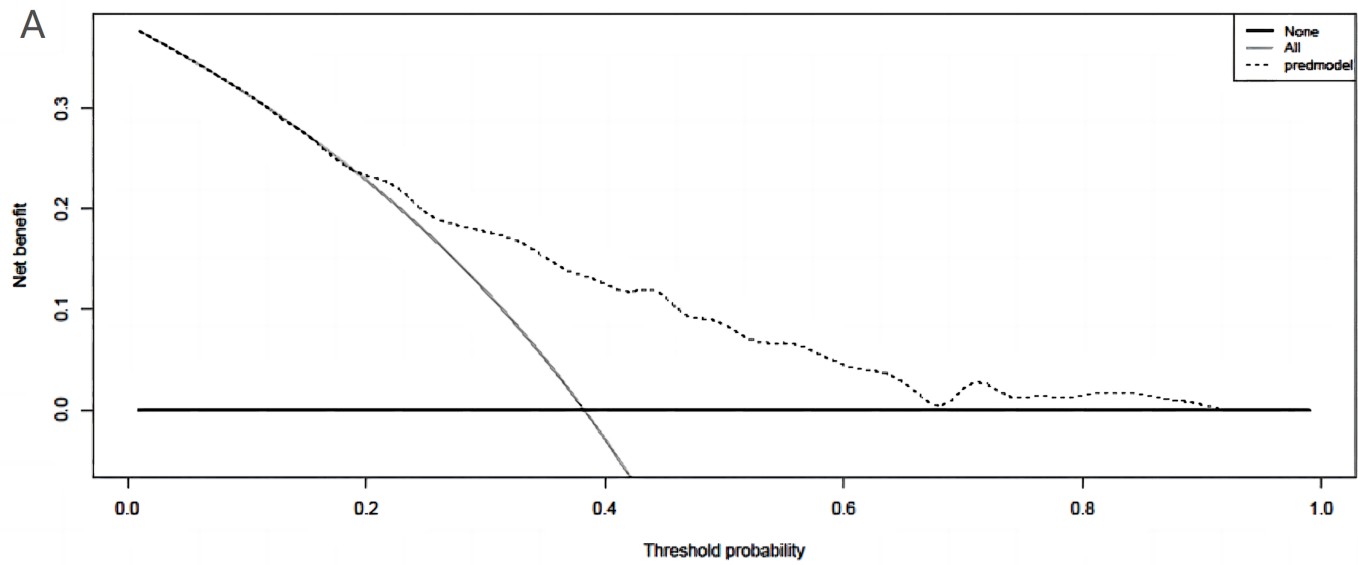

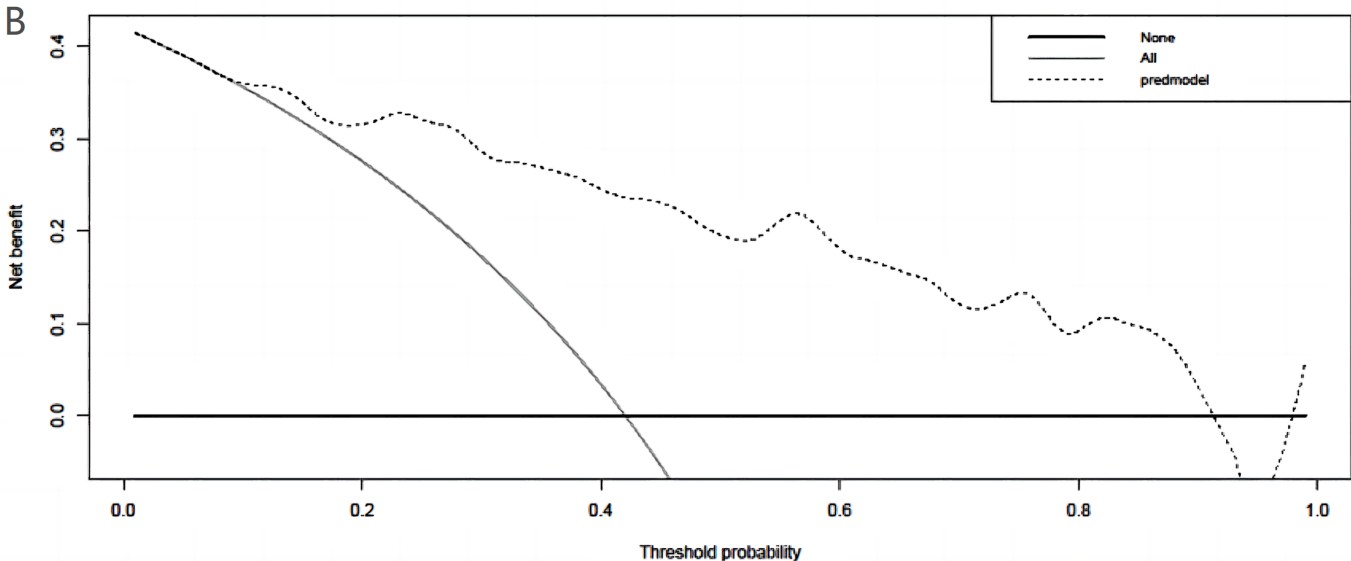

**Figure 5 Clinical decision curve analysis (DCA) for the development set and validation set column line plots.** (A) DCA of column-line plots of predicted risk of myelosuppression in the development set; (B) DCA of column-line plots of predicted risk of myelosuppression in the validation set. The black dashed line is the predictive model; the solid gray line is all patients who were intervened, and the solid black horizontal line is all patients who were not intervened. The graph depicts the expected net benefit per patient in predicting the risk of myelosuppression formation relative to the nomogram. The net benefit increases as the model curve is extended. (Use bootstrap resampling (times = 1,000).)

utilized indicators that do not impose additional financial burden on patients. The resulting model identified age, Ann Arbor stage, number of chemotherapy cycles, surgical history, and neutrophil levels as significant predictors of myelosuppression in this patient population. A number of studies have reported an increased risk and severity of chemotherapy-induced myelosuppression with age. The present study utilized a cut-off point of 60 years of age, as it is commonly regarded as the threshold for categorizing individuals as older or younger in current research. Hence, age was stratified into subgroups of <60 and ≥60 based on this cut-off value. Our results revealed that age serves as a risk factor for chemotherapy-induced myelosuppression, which aligns with findings from numerous prior studies investigating myelosuppression following chemotherapy in solid tumors (*Furlanetto et al., 2019*; *Wang et al., 2017*; *Li et al., 2023*). On the one hand, the bone marrow reserve decreases with age, leading to reduced tolerance to cytotoxic injury and an increased risk of myelosuppression (*Dong et al., 2023*; *Furlanetto et al., 2019*; *Pan et al., 2019*; *Villa et al., 2019*). On the other hand, the reduction of pluripotent hematopoietic stem cells (PHSC) and their response to hematopoietic cytokines, as well as from the suppression of hematopoiesis by inflammatory cytokines (*e.g.*, interleukin 6 (IL-6) and tumor necrosis factor (TNF) may also increase the risk of myelosuppression (*Li et al., 2023*)). Furthermore, the Ann Arbor stage was identified as a risk factor for myelosuppression in patients with DLBCL following initial chemotherapy, with higher stages indicating distant metastases involving various organs such as the bone, liver, brain, and malignant pleural effusion. Additionally, patients with advanced stages typically exhibit poorer physical function and are at a higher risk for severe myelosuppression post-chemotherapy (*Li et al., 2023*; *Zhang et al., 2018*). Consequently, our study determined that the Ann Arbor stage was a prominent predictor of myelosuppression in this patient population.

In line with previous research (*Nian et al., 2022*; *Zhou et al., 2020*), the present study also suggested that the progressive escalation of myelotoxicity in oncology patients undergoing multiple cycles of chemotherapy can result in immune system alterations. Therefore, prolonged exposure to chemotherapeutic agents and the diminished bone marrow reserve capacity substantially increase the risk of myelosuppression. Hence, the number of cycles of chemotherapy was incorporated into our analysis, which was identified as a significant risk factor. In the present study, DLBCL patients who had undergone surgical procedures within the past 12 months were at an increased risk of developing myelosuppression following chemotherapy. Nevertheless, the underlying cause of this association remains uncertain. The physiological stress response triggered by recent surgical interventions may result in the release of significant quantities of catecholamines and other inhibitory substances that impede bone marrow hematopoiesis. Additionally, medications administered during surgery, such as anesthetics or antibiotics, could potentially have deleterious effects on bone marrow hematopoiesis. The specific mechanism needs to be further researched and discussed. Neutrophils are also called polymorphonuclear cells and are the most abundant circulating white blood cells in humans, accounting for 50–70% of leukocytes (*Zuñiga-Traslaviña et al., 2017*). Myelosuppression, especially neutropenia, is the most common form of toxicity (*Björn*

*et al., 2018*). From a clinical perspective, numerous scholars have described neutropenia as the primary characteristic of leukopenia; thus, these two terms have been used interchangeably in studies on myelosuppression (*Lederer et al., 2022*). Low levels of serum neutrophils suggest impaired low immune status, and such patients are susceptible to the development of myelosuppression after chemotherapy. Our univariate regression analysis of pre-chemotherapy alterations in white blood cells, red blood cells, hemoglobin, and neutrophils suggests that all of these factors could serve as potential indicators for the onset of myelosuppression. Furthermore, the findings from our multivariate regression analyses underscore the significant predictive capability of pre-chemotherapy changes in neutrophil levels for the development of myelosuppression.

Numerous studies have investigated the risk factors for the occurrence of myelosuppression following initial chemotherapy in patients with solid tumors; however, conflicting results have been observed compared to our own research. For instance, *Dong et al. (2023)* reported that gender and low serum albumin levels were predictive factors for myelosuppression development in a small-sample, single-center study of lung cancer patients undergoing chemotherapy. Decreased nutritional status was correlated with diminished health outcomes, potentially linked to comorbidities, reduced efficacy of adjunct therapies, and an increased risk of myelosuppression at equivalent dosages. Gender-specific variations in hormone levels, as well as unique physiological states such as menstruation, perinatal period, and menopause, may negatively impact the pharmacodynamics and metabolism of chemotherapeutic agents, leading to drug accumulation within the patient's system. Nonetheless, our two-center experimental analysis revealed no significant influence factor. This may be attributed to variations in the criteria used for classifying low protein levels. Additionally, a large number of patients with hypoproteinemia were excluded from the study due to evident contraindications to chemotherapy during the data collection phase, aligning with the perspectives of *Li et al.*'s *(2023)* research team. However, whether gender is a factor influencing the occurrence of myelosuppression requires further exploration in a large-sample, multicenter study. Furthermore, additional research has indicated that risk factors for chemotherapy-induced myelosuppression encompass low BMI and renal function abnormalities, which impact the metabolism of chemotherapeutic agents, leading to myelosuppression. These factors were analyzed in our study. While the factors exhibited significance in the univariate analysis, the multivariate regression analysis showed no statistical significance. We acknowledge that the majority of existing studies on myelosuppression have focused on patients with solid tumors. In contrast, our study specifically targeted patients with diffuse large B-cell lymphoma (DLBCL), a type of hematological malignancy. Compared to solid tumors, hematological malignancies often involve more aggressive chemotherapy regimens that are inherently more myelosuppressive, which may explain the differences in observed risk factors. Additionally, bone marrow involvement is more common in patients with DLBCL, further contributing to a higher baseline risk of myelosuppression regardless of other factors. Therefore, caution should be exercised when comparing our findings with studies involving solid tumors, as the patient populations and treatment regimens differ substantially.

As previously stated, key risk factors were effectively integrated into the final analyses through comprehensive examination. Consequently, our risk prediction model exhibited notable discriminatory and calibration capabilities in both study centers. Furthermore, the reliability of our model and its potential for improved clinical utility have been validated. This study aimed to investigate the development of a clinical predictive model in DLBCL patients to assess the risk of myelosuppression following first-line chemotherapy. The findings of this research may provide valuable insights for physicians and patients in making informed decisions regarding the necessity of adjuvant treatments, such as prophylaxis against colony-stimulating factor (CSF) or other interventions.

Despite the many strengths of the present study, the limitations should also be acknowledged. Firstly, the relatively small sample size from two medical centers may have prevented the identification of a sufficient number of potential risk factors. Second, the design of this study limited the analysis of factors such as chemotherapy regimen, level of serum LDH, and physical status (PS), among others, all of which may have a potentially significant impact on the development of myelosuppression. Given the differences in chemotherapy regimens and patient characteristics between hematological malignancies and solid tumors, future research should aim to develop separate predictive models tailored to each cancer type to improve accuracy and clinical applicability. In addition, the limited sample size may have hindered the discovery of significant differences between groups, thereby affecting the efficacy of our study. Our future studies will further increase the sample size, influencing factors, and study centers to enrich our findings.

## CONCLUSION

In conclusion, our study investigated the incidence of myelosuppression in DLBCL patients following first-line chemotherapy at two oncology centers in Southwest China using both retrospective and prospective cohort studies. Additionally, a nomogram was developed to forecast severe myelosuppression in DLBCL patients after first-line chemotherapy. The nomogram in our research provided an objective and precise estimation of the risk of developing myelosuppression in DLBCL patients, offering potential clinical utility in predicting and mitigating myelosuppression in this patient population.

### Funding
This work was funded by the Guangxi Natural Science Foundation General Project (No. 2023JJA140882), and the Wu Jieping Medical Foundation Clinical Research Special Fund (No. 320.6750.2024-13-14). The funders had no role in study design, data collection and analysis, decision to publish, or preparation of the manuscript.

## Grant Disclosures

The following grant information was disclosed by the authors:

Guangxi Natural Science Foundation General Project: 2023JJA140882.

Wu Jieping Medical Foundation Clinical Research Special Fund: 320.6750.2024-13-14.

## Competing Interests

The authors declare that they have no competing interests.

## Author Contributions

- Xuexing Wang conceived and designed the experiments, performed the experiments, analyzed the data, prepared figures and/or tables, and approved the final draft.
- Rong Zhang conceived and designed the experiments, performed the experiments, analyzed the data, prepared figures and/or tables, and approved the final draft.
- Haoling Xie conceived and designed the experiments, performed the experiments, analyzed the data, prepared figures and/or tables, and approved the final draft.
- Jinsong Xu conceived and designed the experiments, performed the experiments, analyzed the data, prepared figures and/or tables, and approved the final draft.
- Jie Chu conceived and designed the experiments, performed the experiments, analyzed the data, prepared figures and/or tables, and approved the final draft.
- Chunmei Wei conceived and designed the experiments, performed the experiments, analyzed the data, prepared figures and/or tables, and approved the final draft.
- Quanfang Chen conceived and designed the experiments, performed the experiments, analyzed the data, prepared figures and/or tables, authored or reviewed drafts of the article, and approved the final draft.

## Human Ethics

The following information was supplied relating to ethical approvals (*i.e.*, approving body and any reference numbers):

The research strictly followed the guidelines set forth in the Declaration of Helsinki and obtained approval for its methodology from the Ethics Committee of the Anning First People's Hospital Affiliated with Kunming University of Science and Technology. Informed consent for this retrospective study was waived by the Ethics Committee of the Anning First People's Hospital Affiliated with Kunming University of Science and Technology (Approval No. 2024-035(Self-Horizontal)-01)

## Data Availability

The raw data is available in the Supplemental Files.

## Supplemental Information

Supplemental information for this article can be found online at http://dx.doi.org/10.7717/peerj.19539#supplemental-information.

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
