# Peer review of "Risk factors analysis and nomogram development for myelosuppression in diffuse large B-cell lymphoma patients undergoing first-line chemotherapy: a dual-centre retrospective cohort study"

_PeerJ, doi:10.7717/peerj.19539_

## Round 0.1 · original submission · Major Revisions

Please address concerns of all reviewers and revise manuscript accordingly.

Reviewer 1 ·

Basic reporting

There’s a substantial number of patients and well characterized training and validation cohorts in the manuscript. On the other hand the myelosuppression is not the most frequent complication of R-CHOP-like regimens (apart from R-DA-EPOCH) – so more details about types of R-CHOP like regimens are required.

CTCAE v.5 is more frequently used tool for toxicity evaluation and should be added to (or substitute) WHO chemotherapy toxicity assessment. Grade 2 toxicity is a moderate mylosupression and is transient, so duration of it, need for G-SCF and complication like infections and sepsis as well as delaying next cycle of chemo should be used as myelosuppression indicators rather than only grade. It is not correctly to count each haematological toxicity as myelosuppression event because there’s a big difference between anemia, thrombocytopenia and neutropenia of different grades.

Experimental design

NA

Validity of the findings

Despite interesting analysis the clinical value and application of it is not clear.

Additional comments

I have recommended rejecting this from PeerJ Life & Environment for the reasons above, and also the unlikely scenario that the authors are able to address the comments and queries I have.

Annotated reviews are not available for download in order to protect the identity of reviewers who chose to remain anonymous.

Reviewer 2 ·

Basic reporting

Generally the paper is clear an unambiguous. In line 78-82, I recommend you to rephrase the sentence to express more clearly since the time period when patients were hospitalized at the Anning First People's Hospital seems not correct. In data collection section, you should mention how did you collect the data and how did you deal with missing data. I notice you mentioned these in the statistical analysis part, but you should move them to the data collection section. In statistical analysis section, analyses performed in SPSS and analyses performed in R should be mentioned separately. Line 138-139 should be reorganized. In line 140, p should be smaller than 0.05. It will be confusing if you use development set to represent training set without explanation in any section. Same for validation set, you should not use test set to represent it.

Experimental design

In table 2, around 10 (out of 24) variables have statistically significant differences between the training and validation sets, so patients in the training set are quite different from the patients in the validation set. In this case, your analysis results can be biased. I recommend to use cross validation to analyze all the 350 patients and then the average AUC for the training set and the validation set can be reported. Also, can you explain more how you use univariate logistic regression to selection important features in your predictive model, as well as how did the variables exhibit 5-10 times?

Validity of the findings

The findings are expressed clearly.

Additional comments

To strengthen the introduction, please refer to the articles with PMID: 38911370 and PMID: 38086101 for additional context and supporting evidence.

·

Basic reporting

The text is written in an understandable, fluent language. References are appropriate and sufficiently used. The text is sufficiently suitable in terms of language to be published within its general framework. Also, the tables are sufficient, but some points still need corrections. These will be listed below.

Experimental design

The scientific content of the article is sufficient in terms of its purpose, design and implementation. An original and previously studied subject has been approached in the light of mutually supporting data from 3 different study groups. There is a sufficient number of patients and data.The methodology section also explains these sufficiently. In addition, the explanation of the statistical evaluations is conveyed in a very complicated way. A simple explanation summarizing the statistical data at the end of the paragraphs, along with statistical explanations that even someone who is an expert in the subject may have difficulty understanding, would be beneficial for the readers.

Validity of the findings

Although the data is sufficient, there are some deficiencies. For example, it is easy to explain the relationship between neutropenia before chemotherapy and myelosuppression. However, why is the presence of a surgical history in patients associated with myelosuppression? Which surgeries are in this group, which are listed asmajor surgeries? Is the group with splenectomy among these? Questions such as these have not been answered. In addition, infection-related parameters were not mentioned among the biochemical parameters that examined. We know that infections themselves can also cause suppression in the bone marrow. If there is any data on this subject, their explanations would be useful. The biochemical markers examined should also be stated more clearly in the article. Another point is that the comparisons between solid organ tumors and hematological malignancies are unnecessarily extended. We know that the protocols in solid organ tumors are much different and comparing them with lymphoma protocols does not carry as much meaning as it is thought.

Additional comments

The methodology section of the article (between lines 83-85) states that the patients were diagnosed according to the Hodgkin lymphoma diagnostic criteria. This section, where the abbreviation HL is used, needs to be corrected.

---

## Round 0.2 · Minor Revisions

Please address the remaining issues pointed out by both reviewers and amend the manuscript accordingly.

Reviewer 2 ·

Basic reporting

The grammar in lines 85–89 is incorrect.

Experimental design

Please remove Sections 2.4 and 2.5 and consolidate all relevant information on outcomes and statistical models into the Statistical Analysis section. The authors should specify which statistical models were used to predict the outcome. Additionally, the outcome variable should be clearly defined in this section, including its data type (binary or continuous), the missing data rate, and the approach taken to handle missing values. The discussion on missing data handling should also be moved to the Statistical Analysis section. In Figure 1, you mentioned that several patients were removed due to missing data. Please clarify this process by explaining when missing data imputation was performed and under what circumstances data were removed.

Validity of the findings

Line 214 mentions the test set. Could you clarify what the test set refers to in your study? Additionally, what statistical test was used to compare the ROC curve between the development set and the test set? Please include this information in the paper.

·

Basic reporting

The text is again written in an understandable, fluent language. References are appropriate and sufficiently used. The text is sufficiently suitable in terms of language to be published within its general framework.

Experimental design

The scientific content of the article is sufficient in terms of its purpose, design and implementation. An original and previously studied subject has been approached in the light of mutually supporting data from 3 different study groups. My only concern is that there is still a need for simple explanations summarizing the statistical data at the end of the paragraphs which would be beneficial for the readers.

Validity of the findings

The authors seem to have completed the corrections I have previously mentioned regarding this section sufficiently. There does not seem to be a need for additional corrections.

Additional comments

The authors seem to have completed the corrections I have previously mentioned regarding this section sufficiently. There does not seem to be a need for additional corrections.

---

## Round 0.3 · accepted · Accept

All remaining issues pointed by the reviewers were adequately addressed and revised manuscript is acceptable now.